# Effect of Nano-Sized Energetic Materials (nEMs) on the Performance of Solid Propellants: A Review

**DOI:** 10.3390/nano12010133

**Published:** 2021-12-31

**Authors:** Weiqiang Pang, Chongqing Deng, Huan Li, Luigi T. DeLuca, Dihua Ouyang, Huixiang Xu, Xuezhong Fan

**Affiliations:** 1Xi’an Modern Chemistry Research Institute, Xi’an 710065, China; dengcq2005@163.com (C.D.); lihuan186029@sina.com (H.L.); xhx204@163.com (H.X.); xuezhongfan@126.com (X.F.); 2Science and Technology on Combustion and Explosion Laboratory, Xi’an Modern Chemistry Research Institute, Xi’an 710065, China; 3Space Propulsion Laboratory (SPLab), Politecnico di Milano, I-20156 Milan, Italy; luigi.t.deluca@gmail.com; 4College of Resource Engineering, Xi’an University of Architecture and Technology, Xi’an 710068, China; oydh2013@126.com

**Keywords:** material chemistry, nano-sized energetic materials (nEMs), solid propellant, hazardous properties, mechanical sensitivity

## Abstract

As a hot research topic, nano-scale energetic materials have recently attracted much attention in the fields of propellants and explosives. The preparation of different types of nano-sized energetic materials were carried out, and the effects of nano-sized energetic materials (nEMs) on the properties of solid propellants and explosives were investigated and compared with those of micro-sized ones, placing emphasis on the investigation of the hazardous properties, which could be useable for solid rocket nozzle motor applications. It was found that the nano-sized energetic materials can decrease the impact sensitivity and friction sensitivity of solid propellants and explosives compared with the corresponding micro-sized ones, and the mechanical sensitivities are lower than that of micro-sized particles formulation. Seventy-nine references were enclosed.

## 1. Introduction

As a hot research topic, nano-scale energetic materials (nEMs) have recently attracted much attention in the fields of energy storage and release, catalysis, and adsorption, owing to their unique large specific surface areas (SSA) and extraordinary properties, as the structures, compositions, sizes, and shapes vary on the atomic and molecular scale [1]. In comparison to micro-sized energetic materials (mEMs), the nEMs indicate significantly higher burning rates, lower impact sensitivity, and a higher rate of energy released than those of mEMs [2]. The use of nano technology has provided a great advance in military and industrial applications, especially in micro-electromechanical systems (MEMS) [3], micro thrusters [4], and other applications. For instance, nano-sized ammonium nitrate (AN) and 1,3,5-trimethylene-2,4,6-trinitramine (C_3_H_6_N_6_O_6_, RDX) particles were obtained by a sublimation/condensation process, but a purified explosive sample with a larger scale (i.e., 100 g) is difficult to obtain [5]. Nanostructured 2,4,6,8,10,12-hexanitro-2,4,6,8,10,12-hexaazaisowurtzitane (CL-20) with low sensitivity performance were prepared by solvent/non-solvent cocrystallization technology [6]. The high-temperature treatment method could be used to fabricate nanostructured 1,1-diamino-2,2-dinitroethylene (C_2_H_4_N_4_O_4_, FOX-7) with enhanced sensitivities, as was demonstrated [7]. Quasi-three-dimensional grids of one-dimensional FOX-7 nanostructures have been fabricated via a spray freeze-drying method, which demonstrated size-dependent thermal properties [8]. More interestingly, Liu introduced supramolecular assembly and a de-aggregation process to design and prepare three-dimensional nano hierarchical 2,2,4,4,6,6-hexanitrostilbene (HNS) structures, which possessed favorable thermal stability and safety performance [9]. Results indicated the development of novel energetic materials with nano hierarchical structures is of great significance towards potential applications [10]. However, these methods have disadvantages, such as the complicated processes, long time, and so on [11,12]. This work is focused on the preparation of different types of nEMs, such as nano-sized RDX (nRDX), nano-sized HMX (nHMX)), nano-sized CL-20 (nCL-20), nano-sized TATB (nTATB), nano-sized HATO (nHATO), and nano-sized cocrystals (such as nCL-20/HMX, HMX/HNS, HMX/RDX, etc.). The effects of different nEMs on the properties of solid propellants and explosives were investigated and compared with those of micro-sized ones, placing emphasis on the investigation of the hazardous properties, which could be useable for solid rocket nozzle motor applications.

## 2. Nano-Sized Energetic Materials (nEMs)

### 2.1. Nano Nitramines (nRDX and nHMX)

Compared with ordinary energetic materials, nEMs show some better properties, such as high combustion rate, low mechanical sensitivity, and high energy release rate. When nEMs are applied to solid propellants, the performance of the propellant is expected to be greatly improved [13,14,15]. For example, Fathollahi et al. [16] prepared nano-sized RDX (nRDX) with an average particle size of 180 nm by ball milling, and studied the thermal decomposition characteristics of prepared nRDX. The results show that, compared with the raw material RDX, the decomposition exothermic peak of the nRDX is advanced by 16.8 °C, and the activation energy is reduced by 111.2 kJ·mol^−1^. Liu et al. [17,18] prepared spherical nRDX and nano-sized HMX (nHMX) with average particle sizes of 63.7 nm and 80.3 nm, respectively, by a mechanical crushing method. Compared with micron-sized RDX (mRDX), the friction sensitivity, impact sensitivity, and shock wave sensitivity of nRDX reduced by 30%, 99%, and 59.9%, respectively. The friction sensitivity, impact sensitivity, and shock wave sensitivity of nHMX decreased by 28%, 42.8%, and 56.4%, respectively. Figure 1 shows the SEM images of raw materials and nRDX, nHMX.

Furthermore, nRDX was added to the polymer bonded explosive (PBXs), and the activation energy of PBXs with nRDX decreases by 2.5 kJ·mol^−1^ compared to that of PBX with mRDX, and the impact and friction sensitivity decreases by 55.4% and 21.1%, respectively [22]. For several years, nanostructuring of explosives is a promising field of worldwide research aiming for the goal to desensitize explosives against accidental initiation. By means of the spray flash-evaporation (SFE) technique, nano-sized explosives with reproducible properties were prepared in a single processing step [20].

### 2.2. Nano CL-20 (nCL-20)

2,4,6,8,10,12-hexanitro-2,4,6,8,10,12-hexaazaisowurtzitane (CL-20), as an excellent nitramine explosive with a higher heat of detonation, detonation velocity, and detonation pressure than those of RDX and HMX, can be used in composite modified double base propellants (CMDB) [23], gun propellants [24], and plastic bonded explosives (PBXs) [25]. However, the micron-sized CL-20 (mCL-20) is very sensitive, and explosion would easily happen during production, processing, transportation, etc., which seriously restricted its actual applications. Research has shown that the sensitivities of nitramine explosives are obviously affected by the particle sizes and their distributions [26]. The sensitivities of nitramine explosives are cut down effectively by reducing the particle sizes, and if the nano-sized particles were obtained, the sensitivities would be greatly decreased [27,28]. Nano-sized CL-20 with a semi-spherical shape and 100 nm diameter was prepared using a bi-directional rotation mill method [29], and the results showed that the friction, impact, and shock sensitivities of nCL-20 decreased by 25.0%, 116.2%, and 58.1%, respectively, compared with those of mCL-20.

At the same time, the spherical CL-20 with an average particle size of 200 nm was prepared by ball milling, and its thermal decomposition peak temperature decreased from 245.3 °C to 239.6 °C. Compared with the raw mCL-20, its impact sensitivity and friction sensitivity are reduced by 116.2% and 22%, respectively. A possible reason for this is that nCL-20 is spherical, and has few crystal defects and low porosity, which reduces the probability of hot spot formation [30]. 

Furthermore, nCL-20 with nearly spherical shape and smooth surface particles was prepared with a solution of ethyl acetate through the process solution enhanced dispersion by supercritical fluids, and with the parameters of 30% concentration, 9.0 MPa pressure, 40 °C temperature, 5mL/min solvent flow rate, and 8 kg/h CO_2_ flow rate [31]. Compared with raw mCL-20, the phase transformation temperatures postponed by 30 °C, the thermal decomposition peak temperature advanced by 6.74 °C, and the impact sensitivity significantly reduced.

Additionally, nCL-20 with an average particle size of 100 nm was prepared by a wet mechanical crushing method, and the effect of particle size gradation of mCL-20 and nCL-20 on the properties of CL-20/ 2,4 dinitrobenzene methyl ether (DNAN)/TNT castable explosives was investigated (Figure 2) [32]. When the mass ratio of mCL-20 and nCL-20 is 70:30, compared with the castable explosive with only mCL-20, the impact and friction sensitivity was reduced by 32.7% and 57.1%, respectively. The compressive and tensile strength was increased from 7.93 MPa and 3.48 MPa to 33.74 MPa and 4.94 MPa, respectively. The detonation velocity was increased by 37.0 m·s^−1^. Moreover, nCL-20 was added to the direct writing characteristics of CL-20 explosive ink containing NC/IPA/ethyl acetate by using a direct writing technique, which broadens the research and application direction of nEMs [33].

### 2.3. Nano TATB (nTATB)

1,3,5-triamino-2,4,6-trinitrobenzene (TATB) is a rather insensitive high explosive (IHE), which has good thermal stability and mechanical sensitivity. The energy output of ordinary micro-sized TATB (mTATB) is low, which limits its wide application in future wars. In addition to retaining the excellent properties of ordinary mTATB particles, nTATB also has the characteristics of complete explosion energy release, smaller critical diameter, and more stable detonation wave propagation [5,34,35,36,37]. Based on these advantages of nano-sized performance, nano-sized TATB (nTATB) were fabricated by several methods. For instance, nTATB with an average particle size of 58.1 nm were fabricated using a high-energy milling method (Figure 3) [38], and the results showed that the activation energy of TATB decreased by 13.2 kJ·mol^−1^ after milling, and the 5 s explosion point of nTATB was higher than that of mTATB, meaning that nTATB had a greater thermal reactivity and stability than that of mTATB. 

At the same time, nTATB with a mean particle size of 60 nm and SSA of 31.6 m^2^·g^−1^ was prepared with an alkali-acid recrystallization method and spraying recrystallization method under the condition of volume ratio 1:50 and pH = 7, respectively. The particle sizes distribution of nTATB prepared by alkali-acid recrystallization is from 50 nm to 100 nm, and the particles become a rule crystal shape. Compared with the raw mTATB, its impact sensitivity decreased from 175 cm to 105 cm, and its thermal stability is reduced by 8.8 °C [39]. nTATB was prepared by spraying recrystallization with controllable particle size and higher purity with trifluoromethanesulfonic acid as solvent, and deionized water as non-solvent. The purity of the sample can reach 98.1% when nTATB was purified with water washing to pH = 7 [40]. The SSA of nTATB prepared by a solvent-nonsolvent method is 24.77 m^2^·g^−1^, and its fractal dimension is higher than that of nano-sized HNS (nHNS) also prepared by a solvent-nonsolvent method [41]. The keel-like nanostructure nTATB with a particle size from 70 nm to 400 nm was prepared by means of a solvent/non-solvent method, and the thermal analyses at different heating rates show that the thermal decomposition peak temperature of keel-like nTATB is 1.54–2.91 °C earlier than that of mTATB, the apparent activation energy (*E*_a_) is increased by 0.29 kJ·mol^−1^, and the sensitivity to thermal stimulation is decreased [42].

Among the preparation methods, the high-pressure ultrasonic crushing method has attracted much attention because it does not bring chemical and mechanical impurities. One instant, high-pressure, ultrasonic breaking method was introduced to study the effects of breaking pressure, TATB concentration, surfactant, and breaking times on the mean particle size and surface area of prepared nTATB [43]. The principle diagram of high-speed impact breaking for particles is shown in Figure 4. It was found that under the breaking conditions of interval mode (TATB concentration 10%, pressure 100 MPa, octylphenol polyoxyethylene ether (surfactant, OP), and breaking 10 times), the mean particle size decreases from 18 um to 530 nm, and the thermal decomposition temperature reduces from 396.6 °C to 392.1 °C.

Either concentrated sulfuric acid or NaOH aqueous solution will cause serious pollution to the environment. It is an urgent task to choose an environment-friendly green solution. 1-ethyl-3-methylimidazole acetate is a new green and reusable material, and it, combined with dimethyl sulfoxide (DMSO) in a 70:30 in mass ratio, was used as solvent to prepare nTATB [44]. It was found that its activation energy is lower than that of mTATB, showing that TATB thermal stability is decreased after refining. However, the impact sensitivity of nTATB is a little increased compared to that of mTATB, but the friction sensitivity of nTATB is as low as mTATB.

Furthermore, TATB has excellent safety performance. It is very insensitive to accidental stimuli such as shooting, friction, and falling. At present, it is still the only single insensitive explosive approved by the U.S. Department of Energy. It was reported that reducing the particle size of TATB explosives to a nanometer can not only greatly improve its detonation performance and charge strength, but also improve the short pulse impact sensitivity of TATB; these excellent characteristics make it possible to be used as the starting agent of impact plate detonator, and greatly improve the safety of the impact plate detonator [45]. However, the SSA decreases, and the particles self-aggregate and grow up during the storage of nTATB. The process is affected by environmental conditions, such as temperature, humidity, and air pressure. 

The storage stability of nEMs has attracted more and more attention with their deepening researches. One example, the SSA of nTATB, decreased when increasing the storage time [46]. The stability of nTATB at 90 °C, 10% RH, 50% RH, 90% RH, and 200 Pa were investigated [47]. The results show that the SSA of nTATB decreases obviously after the thermal aging at 45 °C, 60 °C, and 71 °C, and the higher the aging temperature, the more obviously the SSA decreases, accompanied by the growth of some crystal particles. After short-term storage (5 days), the nTATB particles grow up, with the size of about 1–3 μm. The long-term stability of nTATB crystal particles is significantly affected by the extreme humidity and hot environment (90 °C, 90% RH). The nTATB particles also grow and form a micro flake structure after aging in a low pressure (200 Pa) environment at 90 °C. Based on the experiment results, the high surface energy of nTATB facilitates TATB molecules overcoming the energy barrier, which results in the TATB molecules’ diffusion and rearrangement on the crystal surface, and, correspondingly, leads to the grow-up of particles.

### 2.4. Nano TATO (nHATO)

1,1′-dihydroxy-5,5′-tetrazolium dihydroxyamine (HATO, also called TKX-50) is a new energetic ionic salt with a high energy, high detonation velocity, low sensitivity, and low toxicity. It has great application prospects in the fields of solid propellants, explosives, weapon systems, and civil blastings [1,14]. One of the advantages of nano-sized HATO (nHATO) is reducing the mechanical sensitivity compared to micron-sized HATO (mHATO). For instance, nHATO was prepared by a mechanical milling method combined with a vacuum freeze-drying method, and the friction sensitivity and impact sensitivity were tested, which were compared to that of industrial mHATO [48]; the principle diagram of nHATO grinding of prepared nHATO is shown in Figure 5. The results show that the prepared nHATO particles are uniform in size, and spherical in shape, with an average particle size of less than 100 nm. The maximum thermal mass loss temperature is reduced by 2.46 °C (Table 1), the apparent activation energy is decreased by 2.02 kJ·mol^−1^, the self-ignition temperature is increased by 2.95 °C, whereas the activation ΔH, activation entropy ΔS, and Gibbs free energy ΔG are basically the same compared with those of mHATO, indicating that the thermal stability is improved. Moreover, the impact sensitivity and friction sensitivity of nHATO are 44.5 cm and 48%, respectively, which are 44% and 16% lower than those of industrial m HATO (Table 2), indicating that the mechanical sensitivity is significantly reduced, and the safety is improved.

Furthermore, the prepared nHATO was added to PBXs mixed explosives by a solution aqueous suspension method [49]. The results showed that the apparent activation energy decreased by 2.71 kJ·mol^−1^, and the spontaneous combustion temperature increased by 2.95 K under the conditions of a solid–liquid ratio of 1:4, stirring speed of 500 r/min, reaction temperature of 65 °C, and binder concentration of 5%. As another example, nHATO was prepared by means of a spray-freezing into liquid (SFL) method, and the freeze-drying method (FD) method, and different factors were considered to the effect of the prepared particles [50]. It was found that the morphology of the samples prepared by the SFL method is quite different from that of the samples prepared by the FD method. The samples prepared by the SFL method had fluffy three-dimensional network structures, with small particle size and nano particles on the grid, whereas there was no unified form with large particle size in the sub-micron level in the process of the FD method. Figure 6 shows the SEM of typical particles with SFL and FD methods. 

### 2.5. Nano-Sized FOX-7 (nFOX-7)

1,1-diamino-2,2-dinitroethylene (C_2_H_4_N_4_O_4_, FOX-7) is a kind of insensitive and high energy explosive that was synthesized by the Swedish National Defense Research Institute in 1998 [51]. It has been reported to be a high-energy explosive nitroenamine, and is suggested as a potential replacement for RDX and HMX currently used, due to its higher threshold towards impact and friction sensitivity [52]. Nano-sized FOX-7 (nFOX-7), as a very promising alternative energetic material, may be widely used in solid propellants, explosives, pyrotechnics, and other applications [14,53]. For instance, nFOX-7 with an average grain size of less than 30 nm was prepared by the solvent-nonsolvent method by using N,N-dimethylacetamide (DMAC), N-methylpyrrolidone (NMP), and both mixtures as the solvents [54]. Compared with the micro-sized FOX-7 (mFOX-7), the initial thermal decomposition temperatures of the prepared nFOX-7 increase, the temperature ranges between their two thermal decomposition exothermic peaks become narrow, and the decomposition enthalpy greatly increase. When surfactants of OP-10 and Tween 20 were used as the solvents, the initial decomposition temperatures of the prepared nFOX-7 increase by 15 °C, the enthalpies of decomposition increase by 4.97% and 4.65%, respectively, and the friction sensitivities decrease by 66.7% and 50.0%, respectively. 

At the same time, the CoO/FOX-7 nanocomposites were prepared by ultrasonic dispersion process using n-hexane as a dispersion solvent, and the catalytic effects of nano-sized CoO (nCoO) on the properties of prepared nFOX-7 were investigated [55]. It was found that the addition of nCoO reduces the decomposition temperatures of FOX-7. When the addition amount of nCoO was 3.0 wt%, the initial thermal decomposition temperature decreases by 7.47 °C. When the addition amount of nCoO was 4.0 wt%, the second thermal decomposition temperature decreases by 7.14 °C, indicating that nCoO promotes the thermal decomposition of nFOX-7. Another instance, nFOX-7 with an average grain size of 30 nm, was prepared by using a liquid nitrogen ultra-low-temperature recrystallization method under the solvents of N,N-dimethylformamide (DMF), N,N-dimethylacetamide (DMAC), N-methylpyrrolidone (NMP), and acetone (Figure 7) [56]. Compared to the micro-sized FOX-7 (mFOX-7), the temperature difference between the low and high temperature on the decomposition exothermic peaks of prepared nFOX-7 becomes small, and the impact and friction sensitivities significantly reduce. Due to the difference in solubility and polarity of solvents, the particle size of the prepared samples is distinct. The higher the solubility and polarity, the smaller the particle size. When the solvent is N-methylpyrrolidone, the prepared nFOX-7 with particle sizes of below 100 nm accounts for 18%, its initial decomposition peak temperature increases by 11.8 °C, and the thermal decomposition enthalpy increases by 19%. The characteristic drop height increases from 87.2 cm to 138.2 cm, and the friction sensitivity increases from 216 N to 360 N (Table 2). At the same time, the prepared nFOX-7 particles are spherical in shape, and the average particle size of prepared nFOX-7 with NMP as the solvent, and Tween-80 and OP-50 as the surfactants is in the range of 25–40 nm. There was only one exothermic peak in the thermal decomposition process, which was different from the two-step heat release of the raw material, and the initial thermal decomposition peak temperature and enthalpy of decomposition of prepared nFOX-7 increase, and the impact sensitivity greatly reduces. Compared to the mFOX-7, the apparent activation energy of the prepared nFOX-7 calculated by the Kissinger method and the Ozawa method decreased by 22.4% and 31.0%, respectively [57].

### 2.6. Nano-Sized HNS (nHNS)

2,2′,4,4′,6,6′-hexanitrostilbene (HNS) explosive is an insensitive high explosive with good thermal stability. Nano-sized HNS (nHNS) is less sensitive to impact stimuli than that of micro-sized HNS (mHNS). As one example, nHNS with high purity was prepared by the combination of solution quench recrystallization and a solvent-antisolvent recrystallization method [58]. The results show that the prepared crystal appearance of nHNS is smoother with a particle size from 58.9 nm to 231.6 nm, and the Brunauer–Emmett–Teller (BET) SSA is determined to be 19.27 m^2^·g^−1^. The impact sensitivity of mHNS and nHNS are 19.3 cm and 24.7 cm, respectively, and the minimum initiation energy of mHNS and nHNS are 1.1 J and 0.29 J, respectively, indicating that nHNS is less sensitive to impact, but more sensitive to short impulse shock waves than those of mHNS. 

At the same time, nHNS particles with an average particle size of 94.8 nm were prepared by a high energy ball milling method [59]. The results show that the morphology of prepared nHNS is near spherical, and the particle reveals a normal distribution. The apparent activation energy of the thermal decomposition of nHNS is 12.4 kJ·mol^−1^ higher than that of mHNS, indicating that the activation of nHNS molecules requires higher energy. The 5 s explosion temperature of nHNS is 12.2 °C higher than that of mHNS, meaning that nHNS has lower thermal sensitivity.

Even though nano-sized explosives have attracted wide attention due to their unique performance advantage, the lack of understanding of this type of biological toxicity will limit their industrial application. Based on this aspect, the effect of nHNS, nTATB) and nano-sized 2,6-diamino-3,5-dinitropyrazine-1-oxide (nLLM-105) on the toxicity of RAW264.7 macrophages were investigated [60]. It was found that all three types of nano-explosives could significantly reduce the activity of RAW264.7 cells, indicating that nHNS, nTATB, and nLLM-105 have toxic effects on RAW264.7 cells. The toxicity and mutagenicity effect of nTATB was determined, and it was found that it is no actual toxicity and has no mutagenic effect [61].

Additionally, the fractal analysis, widely used in many areas of modern science [62], was introduced into propellants and explosives research. The fractal characteristics of superfine TATB (SSA is 24.77 m^2^·g^−1^) and HNS (SSA is 11.8 m^2^·g^−1^) were carried out [41,63], and it was found that the fractal dimension of TATB is higher than that of HNS in each similar *p*/*p*_0_ range. In addition to the nEMs described above, a spray flash evaporation (SFE) technique was introduced to reduce the particle size of energetic materials as a submicron scale, for example, nRDX and nHMX with particle sizes of 116 nm and 500 nm, respectively, were achieved [64,65]. Since then, this technique was extended to drugs [66]. For instance, ammonium dinitramide (ADN) is an interesting oxidizer to replace ammonium pechlorate (AP) in the composition of solid propellants, and nano-crystallization of ammonium dinitramide (nADN) was prepared by an SFE process to decrease and particle size, and increase the loading rate [67]. Figure 8 shows the SEM images of particle size distribution and morphology of crystallized ADN particles. It was found that the acicular shape of crude ADN have an average length of 25 mm (Figure 8a), and the average diameters of crystallized ADN in methyl acetate and ethyl acetate are 34 nm and 32 nm, respectively (Figure 8d). Furthermore, from the sensitivity behavior of nADN obtained and compared with those of micro-sized ADN (mADN), it was observed that nADN is less sensitive to impact, with a value measured at 4.5 J, but is more sensitive to electrostatic discharge (ESD), with a value of 996 mJ (the ESD of mADN is 1496 mJ), and the sensitivity to friction stays unchanged (Table 2), indicating that nano-crystallized ADN is relatively safe to be handled. 

## 3. Nano-Sized Cocrystal Energetic Materials (nCEMs)

For high-energy materials, the sensitivity and energy are two prominent and opposite problems. High-energy leads to high sensitivity, and low sensitivity leads to low energy. In the field of energetic materials, cocrystals offer the opportunity to design smart explosives, combining high reactivity with significantly reduced sensitivity, which is nowadays essential for safe manipulation and handling. Cocrystal technology can effectively improve the sensitivity and safety performance of explosives and propellants, such as CL-20/HMX, CL-20/FOX-7, HATO/CL-20, etc., and the mechanical sensitivity of nano-sized energetic materials (nEMs) is lower than that of corresponding mEMs [68,69,70]. If the combination of “nano” and “cocrystal” can be realized by means of mechanical ball milling, the sensitivity of high-energy explosives will be further reduced. For example, nano-sized CL-20/HMX (nCL-20/HMX) cocrystals with an average particle size of 81.6 nm were prepared under the conditions of 0.3 mm diameter milling balls [71]. From the SEM images of nEMs cocrystals, it was shown that the micro morphology of prepared nCL-20/HMX cocrystals is near spherical, and the particle size reveals normal distribution. The height of explosion (*H*_50_) with a 5 kg hammer was 32.62 cm, which is much higher than those of the CL-20, HMX and CL-20/HMX simple mixture. Another example [72] found was that nCL-20/HMX cocrystals prepared under optimal conditions are spherical in shape, and the particle size is 80–250 nm after milling for 120 min. Compared with respective energetic monomers, nCL-20/HMX cocrystals exhibit unique crystal structure and thermal decomposition properties. The mechanical sensitivity of the nCL-20/HMX cocrystal is reduced obviously compared to that of raw HMX, whereas the energy output property is equivalent to that of raw CL-20. Furthermore, CL-20/HMX in a molar ratio of 2:1 was chosen to prepare CL-20/HMX nano-plates by spray flash evaporation [73]. Atomic Force Microscopy coupled with Tip Enhanced Raman Spectroscopy (AFM-TERS) was applied to obtain information about the structure and surface composition of single nano co-crystals.

For the other energetic cocrystals, nano-sized CL-20/TATB (nCL-20/TATB) cocrystals with a mean particle size of 61.3 nm were prepared also by a mechanical milling method [74]. The results showed that the prepared sample was spherical in shape, and the activation energy and rate constant of CL-20/TATB cocrystal thermal decomposition were higher than that of raw materials. The mechanical sensitivity of CL-20/TATB is very low. The drop weight (*H*_50_) of prepared nCL-20/TATB with a 5 kg hammer is higher than 90 cm, and the explosion probability (*P*) is only 6%; however, the thermal sensitivity of nCL-20/TATB was higher than that of raw CL-20 and raw TATB, and its 5 s explosion point was 269.8 °C.

Besides CL-20, HMX, and HNS, HMX and RDX can form cocrystals, and nano-sized HMX/HNS (nHMX/HNS) with an average particle size of 93.2 nm, and nano-sized HMX/RDX (nHMX/RDX) with an average particle size of 250.1 nm were prepared by a mechanical ball milling method, respectively [75,76]. The activation energy of nHMX/HNS thermal decomposition is 328.7 kJ·mol^−1^, which is 45.09 and 125 kJ·mol^−1^ higher than that of the raw materials HMX and HNS, respectively, indicating that HMX/HNS has better thermal stability. The drop height (*H*_50_) of nHMX/HNS is more than 90 cm, the friction sensitivity explosion percentage (*P*) is 8%, and the mechanical sensitivity is lower than that of HMX and HNS, revealing that HMX/HNS has a good safety. The impact sensitivity and friction sensitivity of nHMX/RDX cocrystals are lower than those of raw HMX, raw RDX, and HMX/RDX blends. However, the thermal sensitivity of nHMX/RDX is higher than the mw materials, and its 5 s burst point is 196.4 °C. Furthermore, nano-sized HMX/TNT (nHMX/TNT), and nano-sized CL-20/TNT (nCL-20/TNT) cocrystals were prepared by a spray-drying method, and nano-sized nHMX/TATB and nano-sized CL-20/HMX (nCL-20/HMX) were prepared by a mechanical ball milling method, respectively [77]. It was found that the prepared nCEMs mentioned above are spherical in shape, and the particle size of nHMX/TNT, nCL-20/TNT, nCL-20/HMX, and nHMX/TATB cocrystals are 50–200 nm, 50–500 nm, 80–250 nm, and 82–435 nm, respectively. The impact and friction sensitivity of nCEMs reduce obviously compared with raw materials. 

Additionally, nano-sized CL-20/nitroguanidine (NQ) energetic cocrystals (nCL-20/NQ) were prepared by a vacuum freeze-drying method [78]. The schematic diagram for the vacuum freeze-drying process is illustrated in Figure 9. As shown in Figure 10 in the SEM of nCL-20/NQ, NQ represents needle crystal or crystalline powder, and ε-CL-20 particles are irregular polyhedrons with an uneven size and a very wide distribution range. nCL-20/NQ are regularly spherical shaped particles with a narrow size distribution, and most of the particle sizes are less than 500 nm. The result of the mechanical sensitivity test indicated the sensitivity was effectively reduced compared to neat CL-20. According to the hot spot theory, the spherical shape will help to uniformly spread heat to the interior of the particle, and to form fewer hot spots. Therefore, the mechanical sensitivity can be greatly reduced.

**Figure 9 nanomaterials-12-00133-f009:**
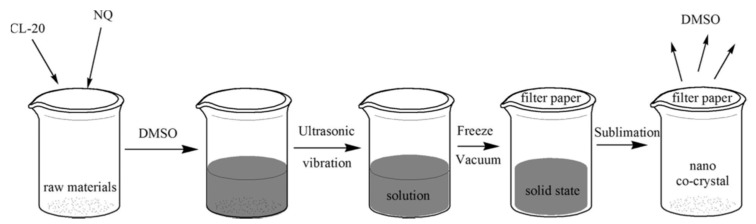
Illustration of the vacuum freeze-drying method to prepare nano-sized CL-20/NQ co-crystal. Reproduced from [78], with permission from Propellants Explos. Pyrotech., 2017.

**Figure 10 nanomaterials-12-00133-f010:**
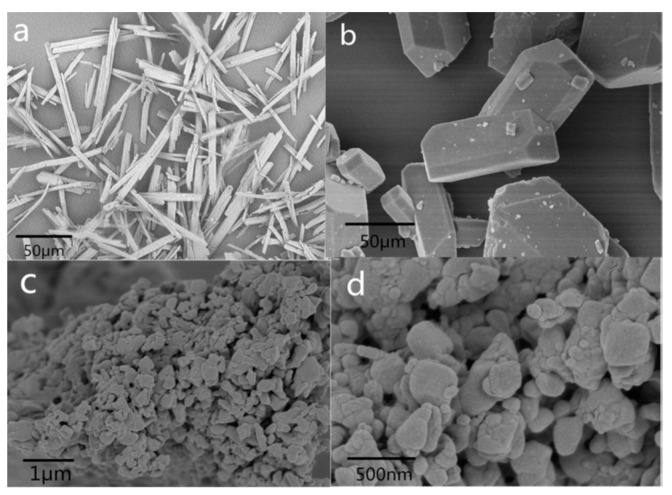
SEM of (**a**) NQ, (**b**) CL-20, and (**c**,**d**) CL-20/NQ nano cocrystals. Reproduced from [78], with permission from Propellants Explos. Pyrotech., 2017.

**Table 2 nanomaterials-12-00133-t002:** Mechanical sensitivity of different EMs compared to that of corresponding micro-sized EMs.

Sample		*H*_50_/cm	*P*/%
mRDX		21.5	88
nRDX		26.1	68
mHMX		23.2	74
nHMX		27.6	57
mCl-20		13.4, 15.0 [67], 11.3 [70]	92
nCl-20		21.8	74
mHATO		31.6	64
nHATO		45.5	48
mFOX-7		87.2 [56]	216 N [56]
nFOX-7	N,N-dimethylformamide (DMF)	123.1 [56]	325 N [56]
	N,N-dimethylacetamide (DMAC)	109.7 [56]	298 N [56]
	N-methylpyrrolidone (NMP)	138.2 [56]	360 N [56]
	Acetone	97.8 [56]	250 N [56]
mHNS		19.3 [58], 52.5 [75]	1.1 J, 46 [75]
nHNS		24.7 [58]	0.3 J
CL-20/HMX mixture		20.1 [71]	-
nCL-20/HMX cocrystal		32.6 [71]	-
TATB		>90 [74]	22 [74]
nCL-20/TATB cocrystal		>90 [74]	6 [74]
nHMX/HNS cosrystal		>90 [75]	8 [75]
HMX/RDX blend		35.7 [76]	85 [76]
nHMX/RDX cocrystal		45.0 [76]	60 [76]
mTATB		170 [43]	0 [43]
nTATB		125 [43]	0 [43]
mADN		3 J [67]	>360 N [67]
nADN		4.5 J [67]	>360 N [67]

In conclusion, with the development of nano-sized materials and explosives, researchers worldwide have developed a variety of nano-sized energetic materials (nEMs) for energetic propulsions. Table 3 lists the summary of advantages and disadvantages of nEMs on the performance of propellants and explosives. Furthermore, the control of the combustion and the detonation properties of high explosives is an important challenge for pyrotechnical science because most applications require energetic devices with a well-defined reactivity. Until now, the only solution to tune the explosive reactivity was to mix several chemicals in order to obtain a composition with the right properties [79].

## 4. Outlook and Perspectives

Over the recent decades, an increasing demand for energetic nano and submicron particles could be observed worldwide. The promising effects of nanostructured energetic materials, such as less sensitivity to external stimuli, and higher performance at the same time, encouraged many research groups worldwide to develop new manufacturing methods. One new type of functional material, nano-sized energetic materials (nEMs), have wide application aspects in various explosives and propulsion systems, including advanced components of propellant compositions, effective energetic improvements of solid propellants and explosives, etc. For example, the addition of nEMs is beneficial to improve the hazardous performance, and possibly reduce the mechanical sensitivities, of propellants and explosives. The impact sensitivity and friction sensitivity of energetic cocrystals are lower than those of raw materials. Nano-sized energetic materials (nEMs) possess the characteristics of high-energy density, large SSA, and a high combustion rate, and have shown good application potential in many aspects, such as MEMS devices, anti-infrared decoy materials, micro thrusters, and high-energy additives. Within current rocket propulsion technology, mostly theoretical and laboratory level applications of nEMs are reported and often for scientific purposes, so much work is needed for their applications at an industrial and practical application level. A number of practical reasons prevent applications at the industrial level, such as particles coating with inert materials, cost, aging, etc. A good control of particle size, morphology, and dispersion is a crucial requirement for applications of nEMs in propulsion.

## Figures and Tables

**Figure 1 nanomaterials-12-00133-f001:**
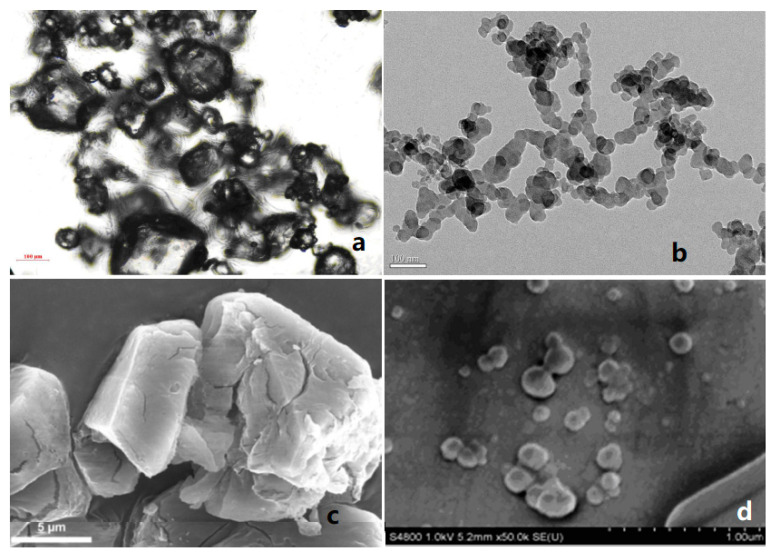
SEM images of micro-sized nitramines and corresponding nano-sized nitramines. (**a**) Raw RDX; (**b**) nRDX; (**c**) Raw HMX; (**d**) nHMX. Reproduced from [19,20,21], with permission from Explosive Materials (Bao Po Qi Cai), 2013; Propellants Explos. Pyrotech., 2014; Explosive Materials (Bao Po Qi Cai), 2014, respectively.

**Figure 2 nanomaterials-12-00133-f002:**
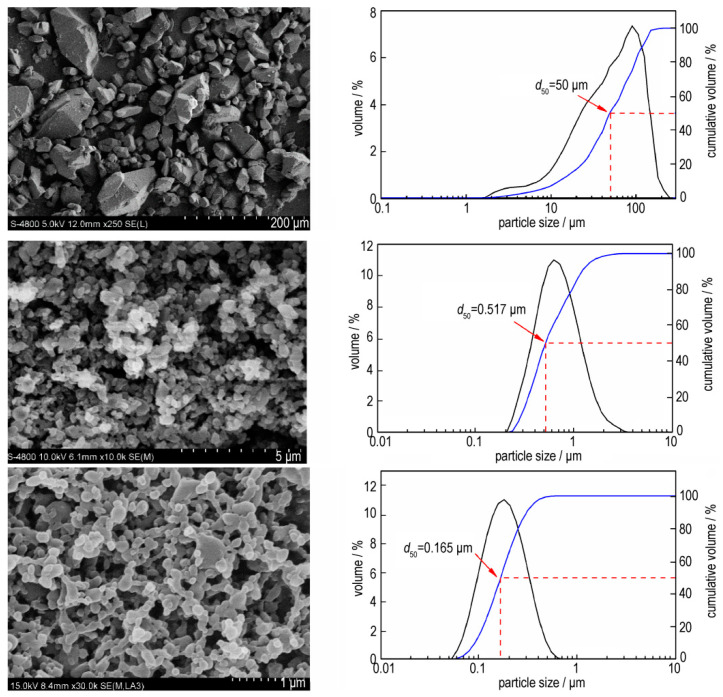
SEM images and particle size distributions of CL-20 with different particle sizes. Reproduced from [32], with permission from Chinese Journal of Energetic Materials (Han Neng Cai Liao), 2019.

**Figure 3 nanomaterials-12-00133-f003:**
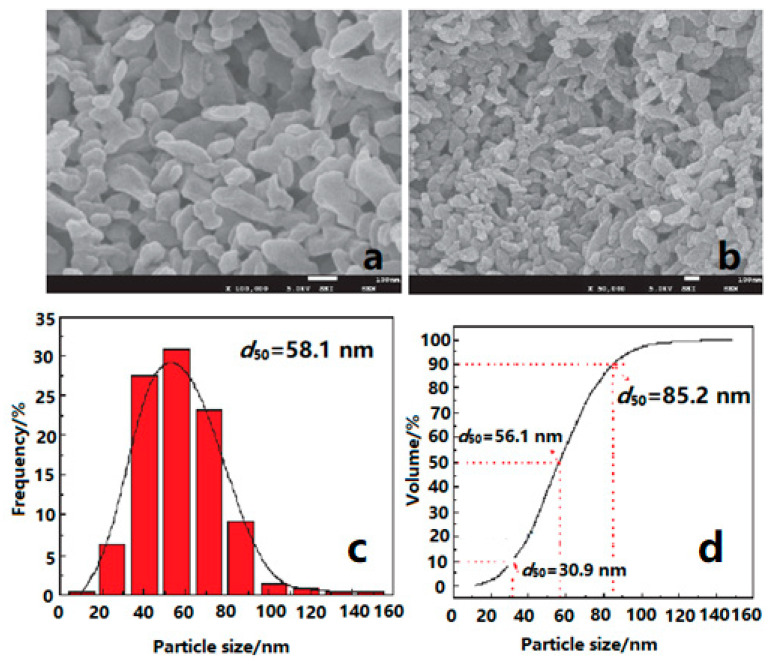
SEM images and particle size distribution of nTATB. (**a**) nTATB, ×100,000; (**b**) nTATB, ×50,000; (**c**) Frequency distribution; (**d**) Cumulative distribution. Reproduced from [38], with permission from Journal of Solid Rocket Technology (Gu Ti Huo Jian Ji Shu), 2017.

**Figure 4 nanomaterials-12-00133-f004:**
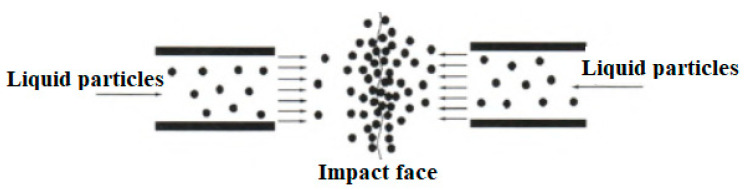
The principle diagram of high-speed impact breaking for particles. Reproduced from [43], with permission from Chinese Journal of Energetic Materials (Han Neng Cai Liao), 2015.

**Figure 5 nanomaterials-12-00133-f005:**
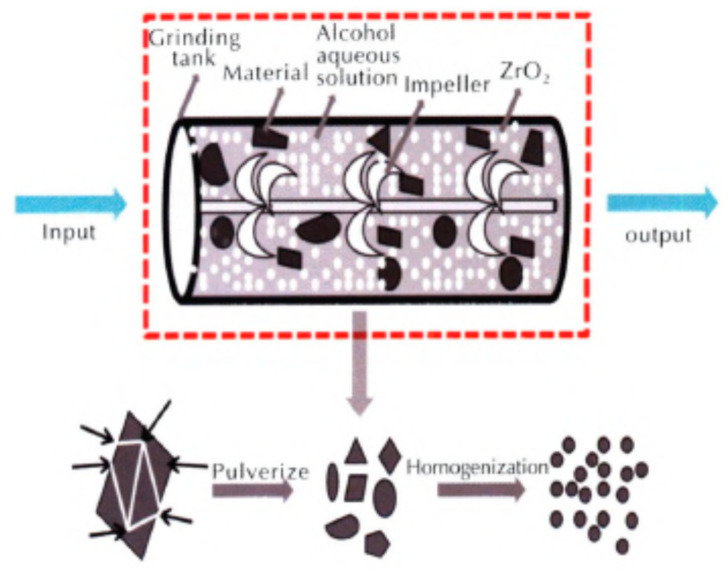
Principle diagram of nHATO grinding. Reproduced from [48], with permission from Chinese Journal of Explosives and Propellants (Huo Zha Yao Xue Bao), 2019.

**Figure 6 nanomaterials-12-00133-f006:**
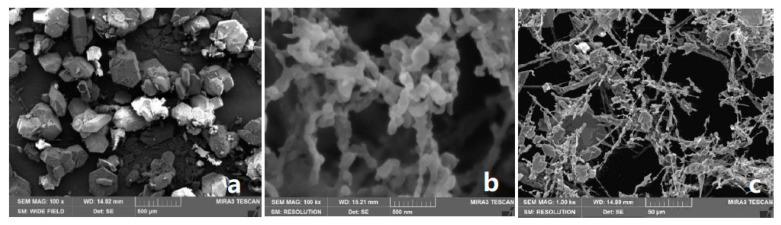
SEM images of different particles. (**a**) Raw mHATO particles; (**b**) nHATO prepared with SFL method; (**c**) nHATO prepared with FD method. Reproduced from [50], with permission from North University of China: Taiyuan, China, 2020.

**Figure 7 nanomaterials-12-00133-f007:**
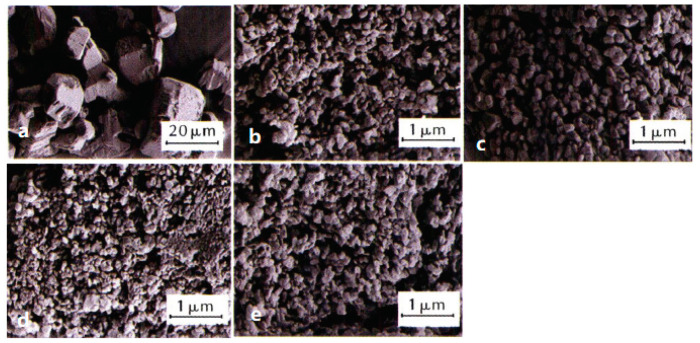
SEM images and average particle size of raw FOX-7 and nFOX-7 prepared in different solvents. (**a**) Raw FOX-7, *d*_50_ = 27 μm; (**b**) DMAC, *d*_50_ = 0.18 μm; (**c**) acetone, *d*_50_ = 0.20 μm; (**d**) NMP, *d*_50_ = 0.14 μm; (**e**) DMF, *d*_50_ = 0.17 μm. Reproduced from [56], with permission from Chinese Journal of Explosives & Propellants (Huo Zha Yao Xue Bao), 2021.

**Figure 8 nanomaterials-12-00133-f008:**
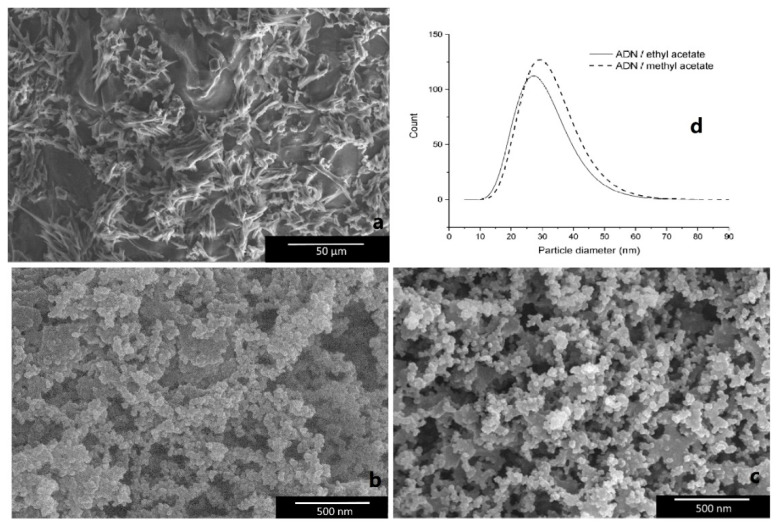
SEM images of raw ADN and nADN. (**a**) Raw ADN; (**b**) ADN crystallized in ethyl acetate; (**c**) ADN crystallized in methyl acetate; (**d**) log-normal size distribution of ADN crystallized in methyl acetate and ethyl acetate. Reproduced from [67], with permission from Propellants, Explosives, Pyrotechnics, 2012.

**Table 1 nanomaterials-12-00133-t001:** Properties of nHATO and mHATO.

Sample	*T*_p_/°C	*E*_a_/kJ·mol^−1^			*T*_o_/°C	*T*_b_/°C
		Kissinger	Starink	Ozawa		
mHATO	246.78	117.88	117.92	120.26	205.95	209.01
nHATO	244.32	115.89	118.33	115.79	208.76	211.96

Note: *T*_p_—peak temperature, °C; *E*_a_—apparent activation energy, kJ·mol^−1^; *T*_o_—thermal decomposition peak temperature when heating rate tends to zero, °C; *T*_b_—spontaneous ignition temperature, °C; *H*_50_—impact sensitivity, cm; *P—*friction sensitivity, %.

**Table 3 nanomaterials-12-00133-t003:** Summary of advantages and disadvantages of nEMs on the performance of propellants and explosives.

Types of nEMs		Advantages and Disadvantages	Refs.
Nano-sized energetic materials	nRDX	The decomposition exothermic peak of the nano RDX is advanced, and the activation energy is reduced.	[17,18]
	nHMX	The impact sensitivity and shock wave sensitivity decreased.	[21]
	nCl-20	The friction, impact, and shock sensitivities of nCL-20 decreased. The phase transformation temperatures postponed, the thermal decomposition peak temperature advanced, and the impact sensitivity significantly reduced.	[29,30,31]
	nTATB	The activation energy of TATB decreased, the 5 s explosion point increased. The activation energy is lower than that of mTATB.	[38,44]
	nHATO	The maximum thermal mass loss temperature is reduced, the apparent activation energy decreased, the self-ignition temperature increased.	[48]
	nFOX-7	The initial thermal decomposition temperatures increased, the apparent activation energy of nFOX-7 calculated by the Kissinger method decreased.	[54,57]
	nHNS	nHNS is less sensitive to impact, but more sensitive to short impulse shock waves than that of mHNS. The apparent activation energy of thermal decomposition of nHNS is higher than that of mHNS.	[58,59]
Nano-sized energetic cocrystals	nCL-20/HMX cocrystal	The mechanical sensitivity of the nCL-20/HMX cocrystal is reduced obviously compared to that of raw HMX, whereas the energy output property is equivalent to that of raw CL-20.	[72]
	nCL-20/TATB cocrystal	The activation energy and rate constant of CL-20/TATB cocrystal thermal decomposition were higher than that of raw materials.	[74]
	nHMX/HNS cosrystal	The activation energy of nHMX/HNS thermal decomposition is higher than that of raw materials. The mechanical sensitivity is lower than that of HMX and HNS.	[75,76]
	nHMX/RDX cocrystal	The impact sensitivity and friction sensitivity of nHMX/RDX cocrystal are lower than those of raw HMX, raw RDX, and HMX/RDX blends. However, the thermal sensitivity of nHMX/RDX is higher than the raw materials.	[77]
	nCL-20/NQ cocrystal	The mechanical sensitivity test indicated the sensitivity was effectively reduced compared to neat CL-20.	[78]

## Data Availability

Not applicable.

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
