# Peer review of "Effect of Nano-Sized Energetic Materials (nEMs) on the Performance of Solid Propellants: A Review"

_nanomaterials, 2021, doi:10.3390/nano12010133_

Round 1
Reviewer 1 Report
I will be brief. I liked the review. It is very useful, it turned out to be a good hand-book, useful for all researchers studying energy-intensive compounds. I confess that after reading the review, I began to believe more in the future of nanomaterials - earlier I was rather skeptical about this direction. I really liked the last phrase in the conclusions “Within the current rocket propulsion technology, mostly theoretical and laboratory level applications of nEMs are reported and often for scientific purposes, much work is needed for their applications at an industrial and practical application level”. Absolutely fair!
Since this is a review, it is natural that those components are described to which the overwhelming majority of articles in the world literature are devoted. True, judging by the links, the authors cite in the overwhelming number of works by Chinese scientists (about half of the links). ...
But I do not exclude that this to a certain extent corresponds to the current contribution of Chinese scientists in the study of energetic materials.In general, the review is very useful.
Author Response
The comments have been revised.

Reviewer 2 Report
Effect of Nano-Sized Energetic Materials (nEMs) on the Performance of Solid Propellants: A Review
The review on Energetic Materials is well organized, so it is judged that it is not unreasonable to publish it in this journal. I would like to make it clear by adding the following simple things.
Characterization data has been sufficiently presented, but data such as textural property as BET surface area should be added and organized.
Author Response
The comments have been revised.

Reviewer 3 Report
The manuscript can be recommended for publication after correcting misprints:
Line 75 - The impact sensitivity and shock wave sensitivity of nHMX decreased by 28%, 42.8% and 56.4%, respectively. – Check numbers.
Line 124 - The detonation speed was increased by 37.0 m·s-1 – In this case, the expression “detonation velocity” is usually used.
Line 298 - Compared to the mFOX-7, the apparent activation energy of the prepared nFOX-7 calculated by Kissinger method decreased by 22.4% and 31.0%, respectively [57]. - Check numbers.
Line 322 - The apparent activation energy of thermal decomposition of nHNS is 12.4 kJ·mol-1, which is higher than that of mHNS, indicating that the activation of nHNS molecules requires a higher energy. - Check numbers.
Line 404 - Most of the particles are 50 mm or more in diameter. - Check numbers.
Lines 464-567 – In references there are 2 references 54 and no reference 53 - Check numbers and references.
Author Response
The comments have been revised.
